# Product Inspection Methodology via Deep Learning: An Overview

**DOI:** 10.3390/s21155039

**Published:** 2021-07-25

**Authors:** Tae-Hyun Kim, Hye-Rin Kim, Yeong-Jun Cho

**Affiliations:** 1Data Science Team, Hyundai Mobis, Seoul 06141, Korea; th@mobis.co.kr (T.-H.K.); hyerin.kim@mobis.co.kr (H.-R.K.); 2Department of Artificial Intelligence Convergence, Chonnam National University, Gwangju 61186, Korea

**Keywords:** defect inspection, deep learning, machine vision, product inspection, smart manufacturing, smart factory

## Abstract

In this study, we present a framework for product quality inspection based on deep learning techniques. First, we categorize several deep learning models that can be applied to product inspection systems. In addition, we explain the steps for building a deep-learning-based inspection system in detail. Second, we address connection schemes that efficiently link deep learning models to product inspection systems. Finally, we propose an effective method that can maintain and enhance a product inspection system according to improvement goals of the existing product inspection systems. The proposed system is observed to possess good system maintenance and stability owing to the proposed methods. All the proposed methods are integrated into a unified framework and we provide detailed explanations of each proposed method. In order to verify the effectiveness of the proposed system, we compare and analyze the performance of the methods in various test scenarios. We expect that our study will provide useful guidelines to readers who desire to implement deep-learning-based systems for product inspection.

## 1. Introduction

Many manufacturing companies apply product inspection systems to detect product defects and to evaluate product quality. The inspection systems examine the possibility of functional problems of the product and determine the location of the defects on the surface of the product. To this end, the inspection system generally uses several camera sensors to examine all or key parts of the products. Some systems that automatically find defects in products based on image-processing technologies can save human effort and labor.

Unfortunately, many conventional methods for defect detection following rule-based algorithms have performed poorly in finding defects in products [1,2]. For example, conventional methods have difficulty dealing with subtle changes in the environment (e.g., small changes in product location or illumination). In addition, they often fail to detect new types of defects owing to their simple criteria. If some defective parts are not detected in the current manufacturing step, they will proceed to the consecutive assembly step and result in significant financial losses. Furthermore, field workers should always be able to manage and adjust the parameters of the system. In this case, the first-time yield (FTY) (The number of good units (i.e., products) produced divided by the number of total units entering the process.) of the products is also reduced.

In order to overcome the limitations of rule-based methods, several methods applying deep learning [3,4,5,6] have been studied in recent years. Deep learning has the following main advantages and strengths: (1) no need for feature engineering; (2) large network capacity to learn from low-level features to high-dimensional representations; and (3) superior performance under various conditions (e.g., illumination changes and noisy images). However, to adopt a deep learning algorithm for a product inspection system, various issues should be considered and are listed as follows:A series of steps to train and utilize deep learning models for the product inspection system in detail;Choosing proper deep learning models for the system;Connecting deep learning models to existing systems;User interface and maintenance schemes.

Although there are many issues, there are only a few studies that have been conducted on how to address these issues and apply deep learning models to the existing product inspection system.

In this work, we efficiently handle all issues based on our knowledge and experience in the real manufacturing field. We aim to build a framework for automatic product inspection based on deep learning techniques and the overall proposed system hierarchy is shown in Figure 1. Edge servers are assigned for each production line and a main server manages each edge server. There are three main stages for applying deep learning to the product inspection system: (1) model training stage, (2) model applying stage, and (3) model managing stage.

In the model training stage, we explain the all steps to train the deep learning models for the product inspection system in Section 3.1. We also provide detailed explanations and guidelines for each step, including data collection, data pre-processing, and choosing proper deep learning models for the system’s purposes.

In the model applying stage, we propose connecting schemes that link the trained deep learning models to the existing inspection system in Section 3.2.1. In general, the majority of equipment such as product inspection systems in old factories are out of date and their resources are limited, e.g., the computing power and storage capacity are insufficient. Therefore, it is difficult to operate deep learning models that require a considerable number of resources in old systems. Instead, we set another workstation as a main server for deep learning and set several workstations as edge servers that are connected with the existing inspection system. To this end, in this study, we propose connection schemes that efficiently link the edge servers with the existing inspection systems to improve the goals of the existing systems.

As a result, deep-learning models can be operated with maximum performance. Moreover, all functions of the existing inspection system such as product management and control units, inspection equipment (e.g., jigs, cameras, and lights), and software can be utilized very stably. Once we successfully trained and applied a deep learning model, we recycled and expanded the trained model to other production lines, as described in Section 3.2.2). This model expansion method reduces the human effort required to train additional deep learning models.

Finally, in the model management stage, we propose an effective model update method that can maintain and enhance the deep learning models of the product inspection system. In order to perform this, we first employed Grad-cam [7], which provides visual explanations from deep networks and helps system managers to understand predictions of deep learning (Section 3.3.1). The proposed model update method includes the model *fine-tuning* and *re-training* processes, as described in Section 3.3.2). The proposed system is fully automated. Moreover, it exhibits outstanding system maintenance, performance, and stability. Owing to the proposed system, even system managers who do not understand deep learning techniques can manage the system very easily.

In order to verify the effectiveness of the proposed system, we compared and analyzed the performance of methods in various test scenarios. Consequently, we can build an automatic product inspection system based on deep learning in a unified framework. The main contributions of this paper can be summarized as follows: (1) This is the first attempt to provide complete steps for building a production quality inspection system based on deep learning, including many helpful guidelines and technical know-how; (2) an introduction to selecting appropriate deep learning models for building product inspection systems; (3) proposing connection schemes that efficiently link deep learning models to existing inspection systems according to system improvement goals; (4) proposing very practical and effective model update methods for system maintenance; and (5) intensive verification of the proposed methods and comparison with other conventional methods.

We hope that our study will provide useful guidelines to readers who desire to implement deep-learning-based systems for product inspection.

## 2. Related Work

### 2.1. Conventional Product Inspection Systems

When visual inspections are conducted by humans, inspection errors may occur owing to factors such as fatigue, inconsistency, and inability to unify the test criteria. In order to reduce human error, many simple methods [1,2,8] have been studied to perform product inspection that automatically determines the location of the defects or classifies the types of defects in the products. They employ simple image-processing technologies, such as image thresholding and binarization [9]. Unfortunately, these methods are too simple to confront subtle changes in the environment (e.g., small changes in product location or illumination) and demonstrate low inspection performance. Recently, some studies [10,11] tried to exploit the conventional image-processing methods for quality inspection of steel and slate slabs. However, it is still difficult to handle the challenges and their applications are quite limited.

In order to overcome the limitations of the simple methods, Chang et al. [12] proposed a more sophisticated method known as the two-phase methodology. In addition, several works [13,14] employed machine learning techniques to propose a high-performance defect detection system. Although machine learning techniques have shown positive results in improving inspection system performance, it is still difficult to manually grasp features that are suitable for defect detection and there are difficulties in terms of generating new models each time a situation where new types of defects occur constantly.

### 2.2. Product Inspection with Deep Learning

With the recent development of deep learning, many attempts to adopt deep learning into product inspection systems have been presented. In particular, not only object recognition and classification studies are conducted [15,16], which are conducted to determine the quality of products based on product image data, but also defect detection technique studies are being performed, which find the location of product defect occurrence.

Studies related to object recognition and classification include the use of AlexNet [17] to recognize defects in dyed fibers or fabrics [4] and the use of VGGNet [18] based model to classify defects on the surface of steel [19]. In addition, a study [6] performed defect detection using sliding window methods to distinguish poor surface conditions (such as scratches and poor junctions). Furthermore, in the case of finding the locations of the defects, there are studies [5,20,21] on the detection of defects on the surface of steel based on yolo series [22,23], variational auto-encoder (VAE) [24], or R-CNN series [25], such as Fast R-CNN [26] and Mask R-CNN [27]. Similarly, [28] applied 3DCNN to analyze three dimensional point cloud data. In  [29], they use RetinaNet [30] to detect the location of the defect and classified the type of defect for the surface of the metal parts. In  [31], the authors present a methodological approach based on the fusion model with two types of the deep networks and random forest models.

In addition, there are papers that help researchers apply deep learning by creating open datasets. Teh NEU surface defect database is a steel plate defect inspection dataset opened by [32] and the aforementioned papers [19,20] also used the dataset. Moreover, KolektorSDD(Kolektor Surface-Defect Dataset) was created by [33] and PCB scans dataset, which are laser scans of PCBs, were generated by [28]. In this paper, we did not use open datasets because we focused on comparing which model is better to use rather than making the state-of-the-art model. However, one of the biggest problems encountered when using deep learning in the defect detection system is the lack of data and, in order to make models work well in industrial applications, pre-training with the open datasets is the best method to improve the quality of the model. Alternatively, there exists study [34] that reduces data usage by applying a few-shot learning method to solve the problem of data shortage.

However, in addition to building an algorithm or model for product inspection, it is necessary to select an appropriate deep learning model that fits the characteristics of the product, connects the deep learning model to the existing system, and to investigate the question of how the design of the new system should proceed so that system managers can manage it without difficulty. However, few studies have been conducted relative to how to address these issues and in applying deep learning models to the existing system. In this study, we propose a new framework that considers all these issues.

## 3. Proposed Automatic Product Inspection Framework via Deep Learning

In order to check the quality of products, systems such as automatic optical inspection (AOI) systems and vision inspection systems have been introduced in the field of manufacturing. In general, these systems use visual sensors, such as RGB cameras or infrared cameras, with various illumination conditions to examine the key parts of the products.

In this section, we explain the overall process of the proposed product inspection system. We categorize the proposed process into three main stages: (1) the deep learning model training stage in Section 3.1; (2) deep learning model applying stage in Section 3.2; and the deep learning model managing stage in Section 3.3. The overall processes of the first and second stages are summarized in Figure 2.

### 3.1. Model Training Stage

#### 3.1.1. Data Collection

It consists of two main parts as follows.

• Image data acquisition. Building a deep-learning model requires a considerable amount of image data. However, in the case of an inspection system, the quantity of data that can be collected depends on various manufacturing conditions, such as production volume, period, and data storage. Since data are generated only when a product is manufactured, it is difficult to create as much data as desired. Thus, the quantity of data is limited. In addition, as images generated by the inspection system generally require a large storage space, there is a limit in the number of images that can be stored. Note that the amount data storage space remaining should be checked regularly and data should either be backed up or more storage space should be added in order to avoid data being deleted.

• Data labeling. After the data acquisition step, the collected data should be labeled as “1’’ (OK: non-defective) or “0’’ (NG: defective) according to its surface condition. We denoted the collected images as $x_{i}$, where *i* is an index of the image and the labels of the *i*th images as $y_{i}$. As the quality and reliability of training data have a substantial effect on the performance of the deep learning model, the labels of the data should be decided very carefully. To this end, many experts such as product quality engineers and deep learning engineers need to collaborate. Potential defects in products are classified in advance according to their type. In addition, evaluation methodologies and policies are clearly established. The collected images are then labeled according to predefined rules for building a training dataset as follows:
(1)$$D=\left\{\left(x_{i},y_{i}\right)|y_{i}\in \left\{0,1\right\},\;i=1,\ldots ,N\right\},$$
where *N* is the total number of collected images.

The steps mentioned previously will not only increase the reliability of models but also provide clear guidelines for system engineers. For example, when an unseen type of defect occurs in the future, we can easily categorize the defect based on the policies.

#### 3.1.2. Data Pre-Processing

Initially, images acquired from the inspection system are usually not optimized and are not sufficient to train deep learning models. In this section, we discuss data pre-processing that refines and augments image data for training deep learning models.

• ROI cropping. Many defect inspection systems contain conveyor belts to transport products and contain many jigs for fixing products to examine their qualities. In general, they consistently fix the products and capture the product images. We call the initial product image the raw image. However, the raw image may include an unnecessary background depending on the inspection system, as shown in Figure 3a. In this case, we need to set the region of interest (ROI) of the manufactured product. The ROI denotes the boundaries of the product in the raw image. We usually set the center position ($c_{x}$, $c_{y}$) and size (*H*, *W*) of the product. Then, ROI can be represented as $b=\left\{c_{x},c_{y},H,W\right\}$.

Specifying the locations of products (i.e., ROIs) is simple but very effective; it rejects many unnecessary regions and allows the analysis methods (Section 3.1.3) to focus entirely on the products, as shown in Figure 3b. Setting ROIs is dependent on the product fixing and localizing accuracy of the inspection system as follows. When products are always located in the same region in the image, we can strictly set the ROIs of the products. Meanwhile, when the product localization is unstable, we have to set ROIs with a loose range so that the cropped image based on the ROIs does not miss the product. Otherwise, one possible solution is to apply object detection methods such as Yolo v3 [22] to set the ROIs under the unstable product position in the raw image.

• Data augmentation.   As mentioned in Section 3.1.1, the amount of data collected depends on the manufacturing conditions. Therefore, a sufficient amount of data for training deep learning models may not be ensured. Unfortunately, deep neural networks commonly require a large amount of training data to learn many network parameters (i.e., weights and bias). In order to tackle the lack of data and to create a training dataset covering various data distributions, we can introduce data augmentation techniques that generate new training data from the existing data. Shorten et al. [35] describes image data augmentation for deep learning and we summarized several possible data augmentation methods in Table 1. There are two categories of data augmentation methods and they are described as follows.

*Image transformation* is a traditional data augmentation method that changes or transforms the given images to augment new image data by the following methods. (1) Perspective transformation changes an image in terms of its size, rotation, and perspective. It has eight degrees of freedom (DOFs) and transforms the image as if the camera observed the images from different viewpoints. (2) Color transformation changes the color distribution of images and color space (e.g., RGB, HSV, etc.) of images. (3) Noise addition adds various kinds of noise to images, such as salt-and-pepper, Gaussian, and Poisson noise. We can add these noises using kernel filtering.*Image generation* creates new images based on the distribution of acquired images. A generative adversarial network [36] was used. Specifically, you can use CycleGAN  [37] or ProgressiveGAN [38]. If you already have multiple defect images on different lines, you can use CycleGAN to create defect images by using good images of the line you want to apply deep learning to(basically good images can be collected very easily). In addition, when images are created using a general GAN, it may be classified as defective by using the form of distortion and this problem can be reduced by using ProgressiveGAN.

In Section 5.1, we present the image classification results before and after data augmentation.

#### 3.1.3. Model Selection and Training: Classifier versus Detector

We expect that the training dataset D for learning deep neural networks will be prepared as described in the previous sections (Section 3.1.1 and Section 3.1.2). It is necessary to select a proper deep learning model to build the product inspection system. The types of products and defects inspected by the product inspection system are very diverse and different types of inspection methods are required for each. In this study, we categorize the deep learning models into two types: defect classifier and defect detector.In this section, we discuss the selection of proper deep learning models according to the characteristics of defects in products.

A common case of product defects is that they appear in the overall area of the product. Figure 4 shows examples of non-defective (OK: 1) and defective (NG: 0) soldered pins. As can be observed, the exteriors of defective soldered pins (NG) have been modified or the lead is completely missing. In such cases, defect classifiers are effective. The distribution of two classes and trained optimal classifiers that distinguish between non-defective and defective parts were compared. Based on a large training dataset D, we can train the weights ($w_{c}$) of a defect classifier. The defect classifier can be represented by a conditional probability distribution as follows:(2)$$p\left(y|x;w_{c}\right),$$
where x is an input image and *y* is the predicted label of the given image x. The classifier $p\left(y|x;w_{c}\right)$ predicts the label of a given image and lies on [0,1]. We can decide the predicted labels of the sample according to probability as follows.
(3)$$y^{pred}=\left\{\begin{array}{c} 1\;\mathrm{if}\;p\left(y|x;w_{c}\right)\geq 0.5,\\ 0\;\mathrm{otherwise}. \end{array}\right.$$

We can utilize several deep neural network models such as ResNet [39], GoogleNet [40], and VGGNet [18] as the backbone network for defect classifiers.

By contrast, many defects would occur partially or locally in the ROI of the product (Figure 5a). For example, foreign matter (e.g., dust, hair, and pollutants) on products and abnormal short circuits are considered as defects. Unfortunately, their locations are not specified but they occur partially or locally on the product, as shown in Figure 5b. The actual defect only accounts for a very small portion of the product’s ROI; therefore, the defect classifier that compares the overall area of products to classify non-defective or defective products is not effective.

In this case, we can employ defect detectors that directly detect the location of defects in the products. In general, the types of foreign matter are not diverse; thus, the defects (e.g., dust, hair, pollutants, and abnormal short circuits) to train defect detectors can be specified easily. By using the training dataset D, we train the weights ($w_{d}$) of a defect detector. In order to train the defect detector, defect positions $\left(b=\left\{c_{x},c_{y},H,W\right\}\right)$ of the training samples in D should be prepared in advance. Refer to Section 3.1.2 and check ROI cropping processes that are exactly the same as the defect positions of each sample. The defect detector can be represented by a conditional probability distribution as follows:(4)$$p\left(b|x;w_{d}\right),$$
where x is an input image and b is the predicted position of a defect. The detector $p\left(b|x;w_{d}\right)$ predicts the defect positions of a given image and lies on [0,1]. If the probability of the detector is larger than 0.8, it can be said that a defect occurs at the position b. When any defect is found, the product (i.e., sample image x) is considered defective. For the defect detectors, Yolo v3, v4 [22,23], and efficient-Det [41] show superior performance in terms of both detection accuracy and speed. As mentioned previously, defect detectors that directly find the location of defects are very effective when there are partial defects on products. In Section 5.2, we show the effectiveness of defect detectors.

Note that we must consider not only the accuracy but also the operating speeds of the models. The accuracy and operating speed of the deep learning model generally follow a trade-off relationship. For example, a model may exhibit good inspection accuracy but requires significant operating time. Then, the model cannot be applied to the inspection system because products should be manufactured within cycle time. This trade-off must be considered before applying deep learning models. We summarized some practical models for defect classification and detection in Table 2.

### 3.2. Model Applying Stage

#### 3.2.1. Connecting Deep Learning Models to Existing Systems

Many product inspection systems in old factories are out of date and their resources, such as computing power and storage capacity, are insufficient. Thus, it is difficult to operate deep learning models that require many resources in old systems. Instead, a possible simple solution is to set another workstation as an edge server for operating deep learning models and to connect the server with the existing inspection system. Since we have separated each system, the deep learning model can operate with maximum performance. Moreover, all functions of the existing system, such as product managing software, control units, and inspection systems including equipment (e.g., jigs, cameras, and lights), can be utilized very stably. In this work, we call the workstation that is running deep learning models the edge server.

According to the current drawback of the existing inspection system, we must clearly define the application purpose of deep learning and the improvement goal of the existing system. The application steps are categorized into three different goals as follows:
Reducing false-positive rates of the existing system;Improving true-positive rates of the existing system;Replacing the existing system with deep learning models.

When false defects (i.e., false-positive rates) occur frequently in the existing system, a deep learning model that prioritizes the reduction in false defects can be applied. This model improves worker productivity and product yield: Workers do not need to perform unnecessary product re-inspections. Meanwhile, the poor defect detection rate (i.e., true-positive rates) of the existing system can be improved by deep learning models to enhance the detection rate. Therefore, the product stability and reliability are improved. In summary, goals 1 and 2 are partial improvements of the system. When both goals are achieved by the trained deep learning model, a complete replacement of the existing inspection system (goal 3) by deep learning will naturally follow. After setting the improvement goals, a connection between the edge server and an existing system is required. In general, the product inspection system mainly consists of three manufacturing parts: (1) machine vision inspection (an existing inspection system), (2) manufacturing execution system (MES), and (3) programmable logic controller (PLC). Please refer to the Appendix A for details on the MES and PLC.According to the improvement goals, connection schemes are divided into two types.

First, when aiming to partially improve the old system (goals 1 and 2), a connection scheme follows the flowchart in Figure 6. In order to reduce false-positive rates (goal 1), an edge server with a deep learning model only predicts the samples confirmed as defective by machine vision inspection. Non-defective samples confirmed by the machine vision are sent to the PLC and MES. Similarly, in order to improve the true-positive rates (goal 2), the edge server only predicts the samples confirmed as non-defective by visual inspection. Defective samples confirmed by the machine vision are sent to the PLC and MES. Note that each goal aims to enhance the drawbacks of existing machine vision systems. When label predictions are performed by an edge server, the edge server directly sends the inspection results to the PLC and MES.

Second, for the complete replacement of the old machine vision system (goal 3), the connection scheme only requires blue arrows in the flowchart in Figure 6. Compared with goals 1 and 2, it is much simpler because it does not consider the prediction results of the old machine vision system, which does not carry out inspection and only records product images to send them to the edge server. Then, the deep learning model inspects the products and sends the results to the PLC and MES. In order to communicate between different systems, we designed a simple socket program in the C# language based on TCP/IP protocols. All terminology used in the flowcharts is summarized in Table 3.

Even when a deep learning model is successfully connected to an old system, it cannot be applied to production immediately. A validation period of at least one month is required to confirm the stability and accuracy of the system and model. During this period, the system should be monitored consistently for any data bottlenecks or abnormal shutdowns.

#### 3.2.2. Model Expansion

If we successfully train a deep learning model for a specific product inspection system, we can consider “model expansion” to improve deep learning models in other inspection systems. In general, mass production plants produce similar types of products in parallel on different lines. However, to train the deep learning models of each production line separately, we need to collect a large amount of data for each inspection system, which requires a significant amount of time and manpower.

In order to address these challenges, we can first leverage a simple transfer learning technique called fine-tuning [42]. Fine-tuning is one of the methods to reduce the amount of data required to learn target inspection models while expanding deep learning inspection systems to other lines. We first set the initial values of the deep learning network with parameters of a trained model, which is already used in another line, and then we perform additional training and parameter modification with the training data obtained from a target inspection system. In this case, it has the advantage of ensuring a certain level of performance, even if there is only a small amount of training data because training does not start from random parameters. However, if the difference between defect types of each line is considerable or there are many environmental gaps, such as illumination or location of parts, it can be difficult to expect simple fine-tuning will result in a great effect.

Domain adaptation can be used to overcome such challenges [43]. This technique adapts two different domain distributions to reduce the discrepancies. This allows the domain distribution of target lines to be adapted to the distribution of a pre-trained deep learning model that is already used in different lines. Specifically, in the research introduced in [44], the domain adaptation technique is effectively used for model expansion in real-world manufacturing lines.

### 3.3. Model Managing Stage

It is highly likely that system managers will lack an understanding of artificial intelligence and deep learning technologies. Since deep learning models are generally not easy to modify intuitively, it is difficult for system managers to maintain and supplement deep learning models. Although conventional product inspection systems can be easily maintained by adjusting a few system parameters, a deep learning model requires much more complex tasks. The system managers need to re-train the deep learning model regularly to render the model robust to unseen data. In addition, they should check whether a proper re-training process has been carried out. In this study, we propose a deep learning model management system aimed at easier and more flexible system maintenance.

#### 3.3.1. Explainable System: Grad-Cam

System managers who do not have any background knowledge of deep learning can have difficulty in managing the product inspection system based on deep learning. The biggest challenge is that they cannot judge whether the deep model is properly trained. In this study, we exploit Grad-cam [7] to alleviate the challenge and use the results to build an efficient model update system.

Grad-cam [7] provides visual explanations from deep networks and highlights the important regions in the image for predicting the labels. Therefore, managers with no deep learning knowledge can easily analyze the Grad-cam results. Figure 7 shows examples of non-defective (OK) and defective (NG) cylinder bonding images with Grad-cam results. The bond should be evenly covered inside the cylinder: if the bond is broken or lumped together, then it is defective. When the deep networks are trained well, the Grad-cam focuses on important regions (i.e., bonding regions) to classify whether the product is defective. Otherwise, Grad-cam focuses on unimportant regions such as background (see Training FAIL cases in Figure 7).

Although the deep model succeeded in classifying the labels, if the Grad-Cam result of the sample is not reliable, then the model cannot cover the sample. In order to render the deep model more robust, system managers collect the image samples with unreliable Grad-Cam and re-train the deep network model with the samples. We explain the proposed model update system in the following Section 3.3.2.

#### 3.3.2. Model Update System

In general, deep learning models perform better than traditional methods, but they are not permanently perfect. For example, the trained model $p\left(y|x;w\right)$ is perfect for the training dataset D; however, there is no guarantee that the model will always be perfect for consecutive testing samples. We expect that the distribution of the test samples is the same as that of the training samples D. Unfortunately, the distribution of the test samples begins to change subtly over a long period of time owing to many factors (e.g., machines becoming obsolete or raw material of products changing). Therefore, continuous maintenance and model updates are required.

In order to update the deep learning model, we can consider two types of update strategies: (1) re-training and (2) fine-tuning [42]. First, re-training re-trains the deep model (i.e., all weights w) from the beginning using the entire training dataset D and new failed samples. It is reliable but requires a large amount of computation owing to the significant training data. Conversely, fine-tuning [42] simply adjusts the trained weights w using only new failed samples. Fine-tuning is very efficient, but it is likely to cause a fatal problem called catastrophic forgetting (Catastrophic forgetting is a phenomenon in which deep learning models forget previously learned information upon learning new information.) [45] when fine-tuning is repeated often. In order to avoid the challenges in updating the deep learning model, we exploit both strategies that complement each other to build an efficient model update system. As shown in Figure 1, we first separated the main server and edge servers for efficiency and stability. The main server is a high-performance workstation (nvidia tesla V100) and performs model *re-training*. The edge servers have nvidia RTX2080 equipped and they perform fine-tuning tasks for each product inspection system maintenance.

• Failed sample collection.    Assume that a trained classifier $p\left(y|x;w\right)$ has tested new samples so that we obtain a set of test samples. Among them, we chose a set of unreliable test samples as follows:(5)$$\begin{array}{cc} D^{u} & =\left\{\left(x_{i},y_{i}^{pred}\right)|0.5-\alpha \leq p\left(y_{i}|x_{i};w\right)\leq 0.5+\alpha \right\},\\ \; & \mathrm{where}\;i=1,\ldots ,N_{t}, \end{array}$$
where $N_{t}$ is the total number of tested samples and $y^{pred}$ is the predicted label of the *i*th sample (refer to Equation (Equation 3)). Note that all testing samples are unseen data ($x_{i}\notin D$). We defined the samples with prediction probabilities around 0.5 as unreliable samples. This is because the samples near the decision boundary (0.5) are not clearly distinguished by the classifier $p\left(y|x;w\right)$. We set α as 0.2 empirically.

There are many uncertainty-based active learning [46] algorithms and we can change the failed sample collection algorithm with them. However, even with this naive sample-collecting logic, we experimentally found that it confirms robustness.Then, the system managers verify the unreliable samples $D^{u}$ by using two-stage verification. To this end, we designed a simple WINDOWS application that demonstrates the tested sample images with two buttons, as shown in Figure 8). In the first stage, system managers verify that the predicted label of the tested sample is correct or incorrect (Figure 8a). If there are incorrect samples, they can build a set of prediction failed samples as follows:
(6)$$D_{f}^{u}=\left\{\left(x_{j},y_{j}^{gt}\right)|D^{u},y_{j}^{pred}\neq y_{j}^{gt}\right\},$$
where *j* is an index of the sample and $y_{j}^{gt}$ is the ground-truth label of the *j*th sample. Although the predicted label of the test sample is correct, the system managers verify the sample one more time. As shown in Figure 8b, they checked the Grad-cam images of samples to verify that the deep learning model predicted the label properly, as explained in Section 3.3.1. According to the result of the Grad-cam, we can also build a poorly trained sample set as follows:(7)$$D_{g}^{u}=\left\{\left(x_{j},y_{j}^{gt}\right)|D^{u},y_{j}^{pred}=y_{j}^{gt},G\left(x_{j}\right)=0\right\},$$
where $G\left(x_{j}\right)$ denotes a Grad-cam verification result for sample $x_{j}$. If the system manager pushed the “Well-trained” button, $G\left(x_{j}\right)$ would be “1”. By contrast, if the “Training FAIL” button was pushed, $G\left(x_{j}\right)$ is assigned as “0”. Thanks toEquation (Equation 7), we consider ambiguous samples as well. Finally, we obtain a set of failed samples during the test as follows.
(8)$$\bar{D}=D_{f}^{u}\cup D_{g}^{u}.$$

It can be represented by $\bar{D}\left(t\right)$, where *t* is a time index. For example, when we set a time slot as 2 days for collecting failed samples, $\bar{D}\left(1\right)$ is a failed sample set collected during the first and second days. Similarly, $\bar{D}\left(2\right)$ denotes the third and fourth days’ failed sample set.

• Model update methods.    A current deep model $p\left(y|x;w\right)$ cannot perfectly classify the samples in $\bar{D}\left(t\right)$. To make the model more accurate, we need to update the model weights w based on $\bar{D}\left(t\right)$. We perform *fine-tuning* to the model on an edge computer (Fine-tuning should be performed when the production line is idle) as follows:(9)$$w^{+}\leftarrow w-\mu \frac{\partial E_{\bar{D}\left(t\right)}}{\partial w},$$
where μ is the learning rate and $E_{\bar{D}\left(t\right)}$ is the loss function of the deep model with respect to the dataset $\bar{D}\left(t\right)$. After the fine-tuning process, the updated model $p\left(y|x;w^{+}\right)$ classifies the failed samples and can better handle upcoming test samples. It is simple but very effective in that it requires a small amount of computation and it significantly improves the model performance through a small adjustment in the model weight distribution.

However, repeating the fine-tuning process often causes catastrophic forgetting [45]. Then, the model performance begins to deteriorate because the model fails to classify the samples trained in the past. In order to prevent the catastrophic forgetting problem, we periodically perform re-train under the following conditions. First, we measure a failed sample ratio as follows:(10)$$\mathrm{FSR}=\frac{\left|{⋃}_{t=1}\bar{D}\left(t\right)\right|}{\left|D\right|},$$
where |·| is a cardinality of a dataset. If the measured FSR is larger than β, we update the training dataset as follows:(11)$$D^{+}\leftarrow D\cup \left(\underset{t=1}{⋃}\bar{D}\left(t\right)\right).$$

This includes not only the original training dataset D but also newly collected samples $\left({⋃}_{t=1}\bar{D}\left(t\right)\right)$ during system testing. Then we re-train the model by the following:(12)$$w^{+}\leftarrow w-\mu \frac{\partial E_{D_{+}}}{\partial w},$$
where $E_{D_{+}}$ is the loss function of the deep model with respect to the new dataset $D_{+}$. The re-training process is performed on the main server because it requires significant computing resources. When model re-training is performed, the model weights are transferred to the edge computer.

Fine-tuning and re-training processes complement one another and they are repeated continuously during the model maintenance. We show several results of the proposed model update system in Section 5.3. The proposed system renders maintaining a model very easy and effective for system managers.

## 4. Datasets

Owing to company confidentiality, it was difficult to open the original images of the data. Instead, we provide details and illustrations of each dataset. According to the types of production lines, we collected PCB parts, Cylinder bonding, and Navigation icon datasets as follows.

The PCB parts dataset contains two types of components in PCB, soldered pins (see Figure 4), and micro-control units (see Figure 5). We collected both defective and non-defective samples for each class, as summarized in Table 4.The Cylinder bonding dataset contains images of cylinders with bond applied. As shown in Figure 7, the bonds should be evenly applied inside the cylinders unless the bonding between the parts will not work properly. If the bond is broken or lumped together, we consider the cylinder as defective. We collected 651 non-defective and 651 defective images.The Navigation icon dataset is a set of icon images in the car navigation software. It includes 20 classes of icons. The size of each icon was normalized to 64×64 pixels. This dataset is insufficient for training deep learning models. In Section 5.1, we present some classification results using the Navigation icon dataset.

## 5. Experimental Results

### 5.1. Data Augmentation

Table 5 shows the results of classification performances before and after the data augmentation process. We tested the Navigation icon dataset and classified 20 different types of common icon images in a mobile application. In order to augment images, we utilized color transformation methods in [47] and produced 10 times as many samples. As can be observed, the classification results after data augmentation improved the classification performance by 22.2%. This implies that data augmentation is highly beneficial. When data are insufficient or unbalanced, we recommend performing simple data augmentation to handle the lack of data problems.

### 5.2. Defect Classification Results

In this section, we address the experimental results on defect classification in terms of datasets: PCB parts and Cylinder bonding.

#### 5.2.1. PCB Parts Dataset

By training the classification networks, we first tested the classification performance of the soldered pins. In order to find the model with the best performance, a total of five networks were used: ResNet [39], vgg16 [18], and GoogleNet [40]. These models were trained in the same environment (e.g., training and test datasets and hyperparameters). ResNet [39] showed the highest accuracy for the classification of soldered pins (Table 6). Based on the results, we used ResNet [39] as the basic model to verify the performance of the other datasets.

In order to find defective MCU in the PCB parts dataset, we trained the defect detection model based on Yolo v3 [22]. Figure 5 shows examples of defects on the MCU part and the defects occur partially. As explained in Section 3.1.3, using a defect detection model is very effective when defects occur partially. Table 7 lists performance comparisons between classification-based models and detection-based model. Yolo v3 [22] showed the highest performance in detecting defective MCU samples. Conversely, the classification-based model, which is used for classifying soldered pins, often fails to detect partial defects in the MCU parts. Based on these experimental results, we verified the proper deep learning models (e.g., defect classifier and defect detector) for each dataset.

#### 5.2.2. Cylinder Bonding Dataset

Adhesive bonding is a process in which a classification algorithm can be applied. Bond is applied to combine two parts and visual inspection is mostly carried out in this process. The errors that occur in this process include over-gluing, insufficient gluing, air bubbles, and slipping [49]. The classification algorithm is more favorable to this process than the detection algorithm, unless the bonding area is wide. This is not only because the inspection duration of classification is normally shorter than detection, but it is also important to see the overall shape of a part for certain types of defects.

As we tested in Section 5.2.1, ResNet [39] showed the best performance for defect classification. Therefore, we employed the deep learning model ResNet [39] as the baseline model for this experiment. In addition, we compared the latest ResNeXt-50, Se-ResNet-50, and Se-ResNeXt-50 algorithms while considering the cycle time of the process. For the used images, we used 20 defective, 20 non-defective for learning, 110 defective, 110 non-defective for testing, and changed the image size to 224×224×3 before comparison. In order to assess the reliability of the test performance assessment, we conducted five rounds of random verification of accuracy and loss. The results are summarized in Table 8. The reason for this result is assumed to be that as this requires focusing on certain defect areas instead of seeing the overall shape and Se-ResNet-50 with attention lines showed a better performance than ResNeXt-50.

### 5.3. User Systems

We first designed an experiment to obtain an experimentally appropriate FSR rate. The experiment was conducted using cylindrical bonding images collected over the past two years. The initial training was performed using 60 images (good: 30; defective: 30) collected in the first two months and the test accuracy was calculated using 262 images (good: 131; defective: 131), where are also obtained within the first 2 months that were not used for training. It was assumed that for every 15 days (one tick in Figure 9), 20 additional images (good: 10; defective: 10) were obtained and used for training. Figure 9 shows the accuracy comparison graph for the existing image according to the number of fine-tunings. We also set fine-tuning parameters to find the right FSR rate, which indicates the frequency of re-training with the entire data. For example, three fine-tuning parameters means that one whole re-training is performed every three fine-tunings, WT only means whole re-training only, and FT only means fine-tuning only.

As can be observed in the graph, there are some dramatic deteriorating points with an accuracy below 90%. When the accuracy was below the expected percentage, it was considered inaccurate, as the system failed to recognize previous images. Thus, a “forgetting problem” is considered to have occurred. For example, 11 ticks with 5 fine-tuning parameters, 40 ticks with 10 fine-tuning parameters, and FSR rates of both points were 0.375 and 0.220, respectively. The forgetting problem does not occur with all points of less than 0.2 FSR rate. When performing fine-tuning, it has been verified experimentally that full re-training must be conducted with a less than 0.2 FSR rate to prevent the forgetting problem and we applied it to the system. If the FSR rate reaches 0.2, the server will perform full re-training. In the experiment, we also confirmed that as |D| increases, the model becomes much more robust; thus, the training *loss* for $\left|{⋃}_{t=1}\bar{D}\left(t\right)\right|$ decreases and even if a large number of $\left|{⋃}_{t=1}\bar{D}\left(t\right)\right|$ are newly trained; however, the deterioration of the model performance appears later. As can be observed in the detailed graph in Figure 9b, all accuracy is deteriorated at 9 tick. It means that there are some changes (e.g., product location or illumination) and something happened with the images. However, even at the tick, the accuracy at 7 fine-tuning parameters is 93.44% and only less than 0.2 FSR rate. We can find out that robustness can be secured by |D|.

## 6. Conclusions

In this study, we have proposed a unified framework for product quality inspection using deep learning techniques. We categorized several deep learning models that can be applied to product inspection systems. In addition, we have studied which deep models were suitable for each system. The steps for building a proposed framework for a product inspection system via deep learning have been explained in detail. In addition, we have addressed several connection schemes for linking deep learning models to existing product inspection systems. Finally, we proposed effective model management methods that efficiently maintain and enhance deep learning models. The results showed good system maintenance and stability.

We have tested and compared the performance of the state-of-the-art methods to verify the effectiveness of the proposed methods in various test scenarios. We expect that our studies will be helpful and will provide guidance for those who want to apply deep learning techniques to product inspection systems.

## Figures and Tables

**Figure 1 sensors-21-05039-f001:**
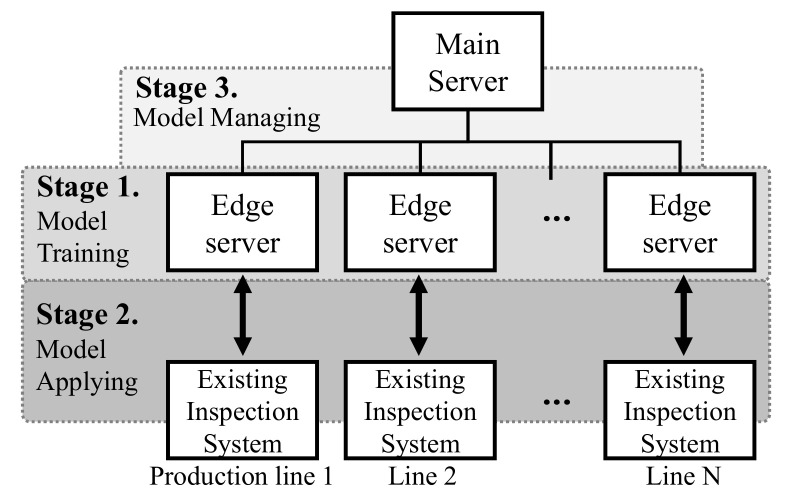
System hierarchy for automatic product inspection via deep learning and proposed three stages for the systems.

**Figure 2 sensors-21-05039-f002:**
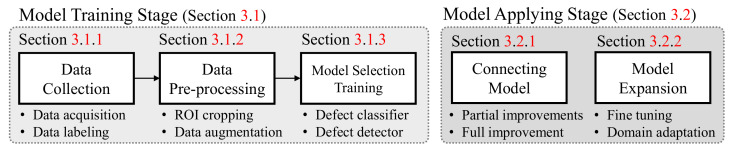
Deep learning model training and applying stages for a product inspection system.

**Figure 3 sensors-21-05039-f003:**
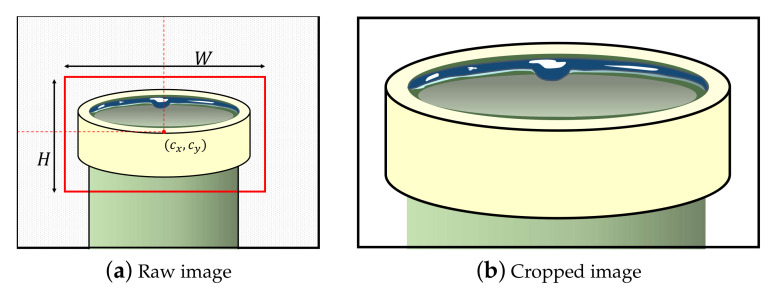
Example of ROI selection: (**a**) raw image, where the shaded area denotes a background region, the solid line denotes a product, and the red box denotes a ROI; (**b**) cropped image according to ROI.

**Figure 4 sensors-21-05039-f004:**
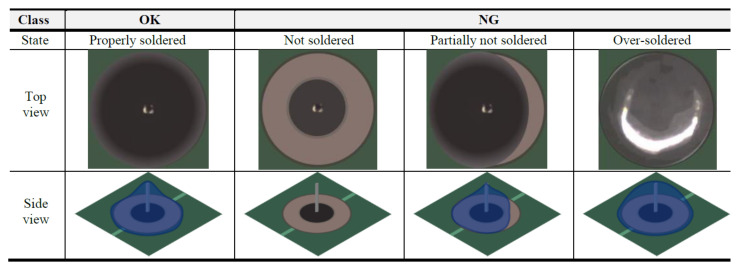
Examples of non-defective (OK) and defective (NG) soldered pin. The blue area in the side view shows the result of soldering. Defects appear in the overall product part.

**Figure 5 sensors-21-05039-f005:**
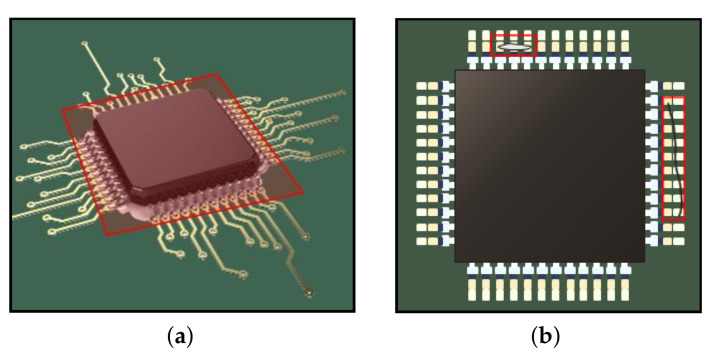
Examples of partial defects on a product. (**a**) Side view: red shaped area denotes ROI. The system finds defects in the ROI. (**b**) Top view: red boxes denote detected defects: abnormal short circuit and hair.

**Figure 6 sensors-21-05039-f006:**
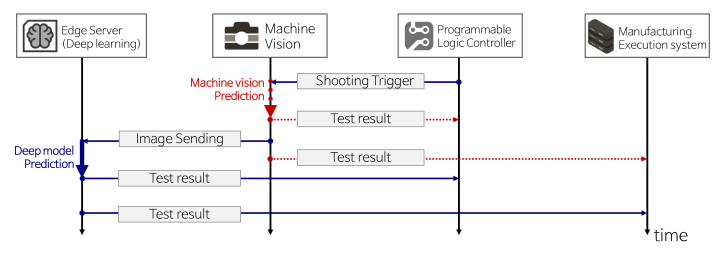
System flowchart to improve the existing system based on deep learning. In order to achieve partial improvements (goals 1 and 2), whole processes (blue arrows and red-dotted arrows) in the flowchart are required. To achieve complete replacement of the old system by deep learning (goal 3), only few processes (blue arrows) are required.

**Figure 7 sensors-21-05039-f007:**
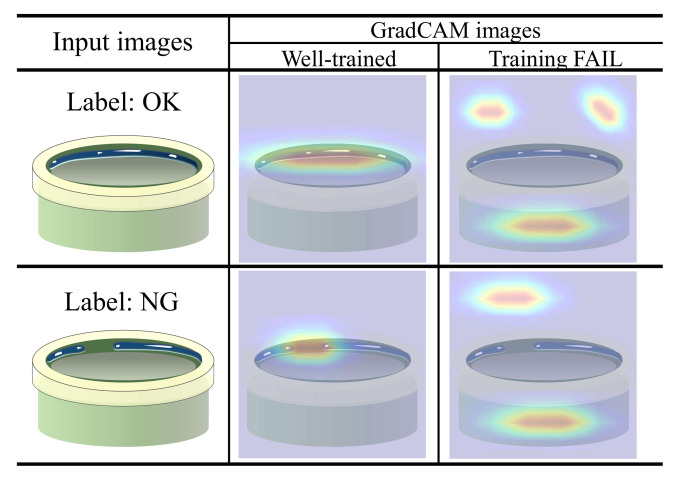
Grad-cam images of cylinder bonding. A warm color indicates high importance.

**Figure 8 sensors-21-05039-f008:**
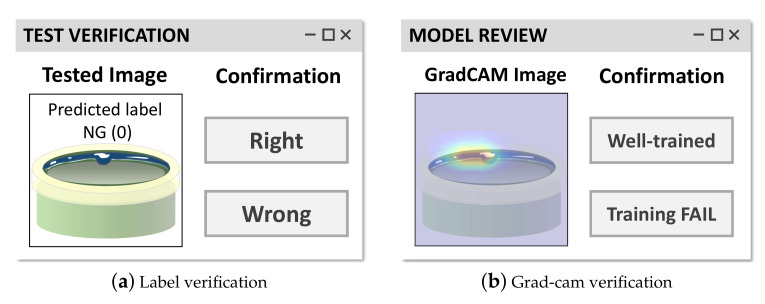
User interface program for verifying unreliable test samples.

**Figure 9 sensors-21-05039-f009:**
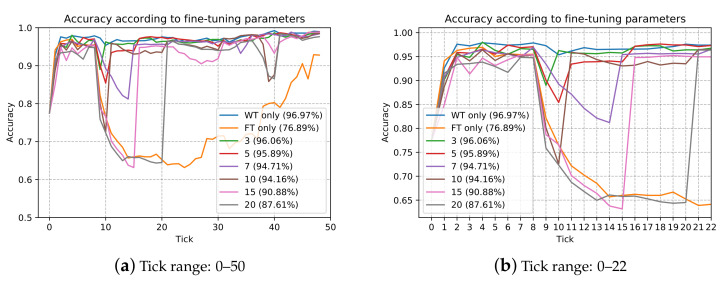
Model accuracy according to update ticks.

**Table 1 sensors-21-05039-t001:** Methods of image data augmentation [35].

Category	Method
Image transformation	Projective transformation
	Color transformation
	Noise addition
Image generation	Generative Adversarial Network [36]

**Table 2 sensors-21-05039-t002:** Differences between classification and detection.

	Defect Classifiers	Defect Detectors
Type of defect	Overall and various types of defects	Local and similar forms
Output	Predict a class (OK/NG) of the image	Detect defects in the images
References	ResNet [39], GoogleNet [40], VggNet [18], and AlexNet [17]	Yolo v3 [22], Yolo v4 [23], and EfficientDet [41]

**Table 3 sensors-21-05039-t003:** Explanation of terminology.

Terminology	Description
Shooting trigger	Requesting the machine vision to start the inspection
Image sending	Sending images taken by machine vision to edge server
Machine vision prediction	Product inspection by machine vision
Deep model prediction	Product inspection by deep learning model
Test result	Inspection result of whether the product is non-defective (OK) or defective (NG)

**Table 4 sensors-21-05039-t004:** PCB parts dataset.

Parts Type	# of Defective	# of Non-Defective
Soldered pin	1000	1000
MCU	2200	2200

**Table 5 sensors-21-05039-t005:** Performance enhancement by using data augmentation.

Class	Number of Images	Accuracy
20	272 (Original data)	0.632%
20	2720 (Original data + augmented data)	0.854%

**Table 6 sensors-21-05039-t006:** Performance comparison of PCB parts: soldered pin dataset.

Methods	Xception [48]	ResNet-50 [39]	vgg16 [18]	vgg19 [18]	GoogleNet [40]
Accuracy	0.954	**0.985**	0.954	0.972	0.963

**Table 7 sensors-21-05039-t007:** Performance comparison of PCB parts: MCU dataset.

	Classification-Based					Detection-Based
**Methods**	**Xception [48]**	**ResNet-50 [39]**	**vgg16 [18]**	**vgg19 [18]**	**GoogleNet [40]**	**Yolo v3 [22]**
Accuracy	0.89	0.83	0.90	0.85	0.85	**0.998**

**Table 8 sensors-21-05039-t008:** Performance comparison of Cylinder bonding.

Method	Average Loss	Mean Accuracy	GFLOPs
ResNet-50 [39]	0.28260	0.941%	3.86
ResNeXt-50 [50]	0.58632	0.915%	4.24
Se-ResNet-50 [51]	0.17950	0.969%	3.87
Se-ResNeXt-50 [51]	0.12076	0.975%	4.25

## Data Availability

Not applicable.

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
