# Peer review of "Product Inspection Methodology via Deep Learning: An Overview"

_sensors, 2021, doi:10.3390/s21155039_

Round 1
Reviewer 1 Report
This paper presents an overview for product inspection methodology via deep learning. This paper is basically well written and significant to this field. More recent articles are suggested to be included with some discussions. For example, "A steel surface defect inspection approach towards smart industrial monitoring," Journal of Intelligent Manufacturing, 2020. "A Steel Surface Defect Recognition Algorithm Based on Improved Deep Learning Network Model Using Feature Visualization and Quality Evaluation," IEEE Access, 2020. "Deep Learning for Smart Industry: Efficient Manufacture Inspection System With Fog Computing," IEEE Transactions on Industrial Informatics, 2018.Author Response
We attach a response sheet for Reviewer #1
We appreciate for your detailed reviews and comments

Reviewer 2 Report
Authors present an overview of deep learning methodologies in quality inspection and propose a framework to cover important aspects of deep learning that are commonly faced in real world industrial production. The proposed framework deals with a multi-stage automatic process in order to successfully apply deep learning methodology in production and continuously adapt to the increasing volume of available data to enhance performance. This work examines the effectiveness of the proposed framework combined with a few of well-known deep learning networks on three datasets. Other datasets that concern quality inspection in industry should also be added in this work for comparison or for reference. Examples are (1) Northeastern University (NEU) surface defect database (https://doi.org/10.1109/tim.2019.2915404), (2) Kolektor Surface-Defect Dataset (https://doi.org/10.1007/s10845-019-01476-x) and (3) PCB_parts (https://dx.doi.org/10.21227/z902-4t15).
This work also presents an overview of deep learning methodologies in quality inspection following real world applications. The related works for both the conventional and deep learning systems should be expanded to provide to the readers a more complete overview of recent works that focus on quality inspection. For the related works on conventional inspection systems, the following recent works can be added:
-
Schmitt, J., Bönig, J., Borggräfe, T., Beitinger, G. and Deuse, J., 2020. Predictive model-based quality inspection using Machine Learning and Edge Cloud Computing. Advanced Engineering Informatics, 45, p.101101.
-
Li, Y., Yu, B., Wang, B., Lee, T. and Banu, M., 2020. Online quality inspection of ultrasonic composite welding by combining artificial intelligence technologies with welding process signatures. Materials & Design, 194, p.108912.
-
Iglesias, C., Martínez, J. and Taboada, J., 2018. Automated vision system for quality inspection of slate slabs. Computers in Industry, 99, pp.119-129.
-
Zhang X, Zhang J, Ma M, Chen Z, Yue S, He T, Xu X. A High Precision Quality Inspection System for Steel Bars Based on Machine Vision. Sensors. 2018; 18(8):2732. https://doi.org/10.3390/s18082732.
The related works with deep learning methodologies should also be expanded, including the following important works:
-
Yun, J. P., Shin, W. C., Koo, G., Kim, M. S., Lee, C., & Lee, S. J. (2020). Automated defect inspection system for metal surfaces based on deep learning and data augmentation. Journal of Manufacturing Systems, 55, 317–324.
-
Dimitriou, N., Leontaris, L., Vafeiadis, T., Ioannidis, D., Wotherspoon, T., Tinker, G., & Tzovaras, D. (2020). Fault diagnosis in microelectronics attachment via deep learning analysis of 3-D laser scans. IEEE Transactions on Industrial Electronics (1982), 67(7), 5748–5757.
-
Block, S. B., da Silva, R. D., Dorini, L. B., & Minetto, R. (2021). Inspection of imprint defects in stamped metal surfaces using deep learning and tracking. IEEE Transactions on Industrial Electronics (1982), 68(5), 4498–4507.
-
Kotsiopoulos, T., Leontaris, L., Dimitriou, N. et al. Deep multi-sensorial data analysis for production monitoring in hard metal industry. Int J Adv Manuf Technol (2020).
-
Han, Y., Fan, J. & Yang, X. A structured light vision sensor for on-line weld bead measurement and weld quality inspection. Int J Adv Manuf Technol 106, 2065–2078 (2020). https://doi.org/10.1007/s00170-019-04450-2
-
Lin, H.-I., & Wibowo, F. S. (2021). Image data assessment approach for deep learning-based metal surface defect-detection systems. IEEE Access: Practical Innovations, Open Solutions, 9, 47621–47638.
-
Benbarrad, T.; Salhaoui, M.; Kenitar, S.B.; Arioua, M. Intelligent Machine Vision Model for Defective Product Inspection Based on Machine Learning. J. Sens. Actuator Netw. 2021, 10, 7. https://doi.org/10.3390/jsan10010007
-
Yang, Y., Pan, L., Ma, J., Yang, R., Zhu, Y., Yang, Y., & Zhang, L. (2020). A High-Performance Deep Learning Algorithm for the Automated Optical Inspection of Laser Welding. Applied Sciences, 10(3), 933. https://doi.org/10.3390/app10030933
Finally, a table to present a comparative summary of the related works on conventional and deep learning inspection methodologies, should be added for completeness of the work.
Author Response
We attach a response sheet for Reviewer #2
We appreciate for your detailed reviews and comments

Reviewer 3 Report
- This paper proposes a product quality monitoring framework based on deep learning technology, explains in detail the steps of building a deep learning-based detection system, and proposes a connection scheme that effectively connects the deep learning model with the existing detection system. The overall structure is relatively complete, but the usage guidelines and technical tips about the entire monitoring system are not very detailed.
- This paper introduces several deep learning models suitable for this detection system, including ResNeT, Se-ResNeT, vgg16, GoogleNet, YoLo v3, etc. However, the comparison is not very detailed. There are many latest deep learning models. Could you do some comparative tests? More defect detection methods are worthy of reference. Triplet-Graph Reasoning Network for Few-Shot Metal Generic Surface Defect Segmentation.
- There are some professional terms in the article that need careful consideration, and the language of the whole article needs to be improved.
Author Response
We attach a response sheet for Reviewer #3
We appreciate for your detailed reviews and comments
